# Brentuximab Vedotin and Pembrolizumab Combination in Patients with Relapsed/Refractory Hodgkin Lymphoma: A Single-Centre Retrospective Analysis

**DOI:** 10.3390/cancers14040982

**Published:** 2022-02-15

**Authors:** Fulvio Massaro, Nathalie Meuleman, Dominique Bron, Marie Vercruyssen, Marie Maerevoet

**Affiliations:** 1Department of Haematology, Institut Jules Bordet (ULB), 1070 Brussels, Belgium; nathalie.meuleman@bordet.be (N.M.); dominique.bron@bordet.be (D.B.); marie.vercruyssen@bordet.be (M.V.); marie.maerevoet@bordet.be (M.M.); 2PhD Program in Clinical and Experimental Medicine, University of Modena and Reggio Emilia, 41121 Modena, Italy

**Keywords:** Hodgkin lymphoma, antibody-drug conjugate, immune checkpoint inhibition, brentuximab vedotin, pembrolizumab, salvage therapy, autologous stem cell transplantation

## Abstract

**Simple Summary:**

The standard treatment for Hodgkin lymphoma (HL) patients presenting a relapsed/refractory (R/R) disease is salvage chemotherapy followed by autologous stem cell transplantation (ASCT). With commonly used chemotherapy combinations, 25–30% fail to proceed to ASCT, with poor outcomes. The aim of this retrospective study was to evaluate the efficacy of brentuximab vedotin (BV) and pembrolizumab combination as a bridge to ASCT in R/R HL patients. We retrospectively collected data from 10 patients, 8 male and 2 female, with a median age of 30.7 years. The median follow-up time was 16.5 months, while the median number of received cycles of treatment was 4 (2–7). Eight patients proceeded to ASCT (80%) and seven of them to subsequent BV maintenance, with two early disease progression (PD). The BV and pembrolizumab combination is a very effective bridge treatment to ASCT for high-risk R/R HL patients.

**Abstract:**

Classical Hodgkin lymphoma (HL) patients presenting a relapsed/refractory (R/R) disease are currently managed with salvage chemotherapy followed by autologous stem cell transplantation (ASCT). However, almost 25–30% of these patients fail to achieve a complete response (CR) with standard salvage regimens. In this retrospective study, we evaluated the efficacy of a combination of brentuximab vedotin (BV) and pembrolizumab in a series of HL patients presenting with a high-risk, multi-refractory disease. Patients achieving a Deauville score ≤4 proceeded to ASCT consolidation. After ASCT, patients received BV as maintenance for a total of 16 administrations. We collected data from 10 patients with a median age of 30.7 years. At a median follow-up of 16.5 months, we reported a complete metabolic remission (CMR) in eight patients (80%), with seven patients (70%) directly proceeding to ASCT (the other two patients in CMR are still undergoing treatment). BV consolidation was started in six patients and completed by three patients (one ongoing, two interruption). Two patients (20%) presented a progressive disease (PD) and subsequently died, while the others are still in CMR. The BV and pembrolizumab combination is a very effective bridge treatment to ASCT for high-risk R/R HL patients.

## 1. Introduction

Classical Hodgkin lymphoma (HL) is nowadays a highly curable disease, with standard first line polychemotherapy regimens achieving a complete remission (CR) rate of 80–90%. However, 20–30% of patients will experience a relapse or a progressive disease (PD) [1,2,3]. The standard approach in this setting is salvage treatment followed by autologous stem cell transplantation (ASCT); almost half of the patients who undergo ASCT present long term disease remissions [4,5]. Several salvage chemotherapy schemes have been tested in this setting and mostly described in retrospective series; therefore, a gold standard treatment has not yet been identified. With commonly used chemotherapy combinations, 25–30% of these patients fail to achieve a complete metabolic response (CMR), which seems to be the most important prognostic factor to achieve a prolonged progression-free survival (PFS) and to proceed to ASCT, with subsequently poor outcomes [6,7].

Single agents brentuximab vedotin (BV) and pembrolizumab have shown efficacy in heavily pretreated HL patients. 

BV is an antibody-conjugated drug which exerts its cytotoxic action towards CD30-positive HL cells. Its mechanism of action is based on the internalization of monomethyl auristatin E (MMAE), a microtubule-disrupting agent, into the target cell, inducing apoptosis as final result. In a phase 2 study conducted on R/R HL patients, BV induced an overall response rate (ORR) of 75% and a CR rate of 34% [8]. 

Pembrolizumab is a humanized monoclonal IgG4 kappa antibody directed against programmed death receptor-1 (PD-1) on lymphocytes. This receptor, which normally prevents the immune system from attacking itself, can be used by tumor cells to escape from anti-cancer immune response. Pembrolizumab monotherapy in R/R HL patients was associated with an ORR of 72% and a CR rate of 28% in the phase 2 KEYNOTE-087 trial [9].

The combination of BV with the PD-1 inhibitor nivolumab has been explored in a phase 1/2 study as first salvage treatment for HL patients; this combination was associated with an ORR of 82% and a CR rate of 61% [10]. 

We performed a retrospective analysis of BV and pembrolizumab combination as salvage treatment in a series of heavily pretreated HL patients. 

## 2. Population and Methods

We retrospectively collected data from ten consecutive HL patients presenting with a high-risk multi-refractory disease (two or more prior treatments), followed at Jules Bordet Institute between May 2019 and October 2021 and treated with a combination of BV and pembrolizumab.

All enrolled patients had biopsy-proven R/R HL and had an ^18^FDG-PET-CT (PET-CT) avid measurable disease. Patients were covered by special insurance conditions permitting treatment administration and reimbursement. Treatment proposal was approved by our multidisciplinary oncology committee and informed consent was obtained from all subjects receiving the salvage combination. 

Treatment consisted of a combination of BV (1.8 mg/kg IV) and pembrolizumab (200 mg IV fixed dose), delivered in 3-week cycles. A PET-CT evaluation was performed after two cycles: patients achieving a Deauville score (DS) ≤4 received high-dose chemotherapy (carmustine, etoposide, cytarabine, melphalan) and ASCT consolidation. Responses were assessed according to the Revised Response Criteria for Malignant Lymphoma [11,12]. Patients could continue to receive further cycles of BV and pembrolizumab before ASCT, at the physician’s discretion. After ASCT, patients received BV as maintenance for a total of 16 administrations (including pre-ASCT cycles). A PET-CT evaluation was performed 90 days after ASCT and at the end of maintenance treatment. 

Toxicity was reported according to Common Terminology Criteria for Adverse Events (CTCAE) score v.5.0.

## 3. Results

We retrospectively analyzed data from 10 patients, 8 males and 2 females, with a median age of 30.7 (20.6–36.4) years. Patients had received a median of 3 (2–5) prior lines of treatment, and the median time from diagnosis to treatment with pembrolizumab and BV combination was 27.7 months. 

Among baseline characteristics, nine (90%) patients presented an advanced disease at relapse, six (60%) a primary refractory disease, six (60%) extranodal disease at relapse and four (40%) a CR duration less than 12 months. Five (50%) patients presented three of these features simultaneously. Table 1 summarizes demographic and baseline characteristics for all enrolled patients. 

All patients completed the salvage treatment and had a PET-CT evaluation. The median follow-up time was 16.5 (2.4–29.9) months, while the median number of cycles was 4 (2–7). 

The ORR was 90%, and particularly, a CMR was achieved from eight patients (80%), with a median time to best response of two cycles. Among the group of responding patients, seven (70%) proceeded to ASCT: in one case, a patient in CMR (DS1) after two cycles presented a DS4 after six cycles (salvage treatment pursued due to ASCT deferral because of the first wave of COVID-19 pandemic) and received consolidative radiotherapy before ASCT. Two patients are scheduled to receive ASCT shortly. 

Stem cell collection was successful for seven out of seven patients, with a median of 5 × 10^6^ CD34+ cells/kg collected. Mobilizing agents included granulocyte colony-stimulating factor (G-CSF) in four cases and G-CSF with plerixafor in three cases. Concerning the other patients, two are expected to be collected in the next weeks while another one was collected before starting the salvage treatment. 

The only patient presenting a PD after two cycles was treated with a combination of nivolumab and gemcitabine, achieving a partial response (PR) and subsequently received radiotherapy and ASCT followed by allogeneic stem cell transplantation (alloSCT) in a tandem strategy. 

BV consolidation was started at a median of 36 (28–85) days after ASCT from six patients and completed by three of them. One patient is undergoing treatment while two others interrupted consolidation (one PD, one consolidation with alloSCT). 

Two patients (20%) presented with PD after ASCT: one patient was in CMR and the other in PR (with a DS5) before ASCT and they relapsed 4 months and 3 months after the procedure, respectively. One patient was treated with a combination of nivolumab and gemcitabine and achieved a CR, while the other was treated with a combination of nivolumab and bendamustine and achieved a PR. Both patients received a consolidation with alloSCT and died from infectious complications (systemic tuberculosis and COVID-19 infection in one case, septic shock in one case). Median progression-free survival (PFS) value was 12.9 months. Table 2 describes in detail the treatment and the outcome of each patient. Figure 1 shows the overall survival (OS) curve. 

Three patients (30%) experienced four adverse events (AEs) during the treatment with pembrolizumab and BV: one case of infusion-related reaction, neutropenia, polyarthritis and hyperthyroidism, respectively, all classified as grade 3 events. Polyarthritis was managed with pembrolizumab discontinuation and systemic corticosteroids treatment, hyperthyroidism with pembrolizumab discontinuation only. Two patients (33%) presented two AEs during BV consolidation: a grade 2 arthralgia and a grade 4 neutropenia. Both cases were successfully managed with BV dose reduction to 1.2 mg/kg. 

## 4. Discussion

To the best of our knowledge, no data are currently available concerning the combination of BV and pembrolizumab in R/R HL. We described the outcomes of 10 patients treated with this combination, reporting a CMR of 80% and a total of 70% of patients proceeding to ASCT directly after this line of treatment. Moreover, response achievement was commonly quick, with most patients in CMR after only two cycles. The safety profile was manageable, with only a few grade 3 events to be mentioned. It has to be underlined that treatment schedule is compatible with outpatient administration.

Herrera and colleagues explored the efficacy of the combination of BV with another PD1 inhibitor, nivolumab; a recently published update reported an ORR of 85% and a CR rate of 67%, with a 3-year PFS of 77% for the entire cohort and 91% for patients undergoing ASCT directly after study treatment. The safety profile was manageable, mainly characterized by grade 1–2 events and 18% of immune related adverse events (IrAE) needing corticosteroids treatment. It is difficult to perform a true comparison among these studies; however, it is interesting to note that in the trial with nivolumab patients were less pretreated (no prior salvage treatments received) and presented less frequently unfavorable disease characteristics (45% of primary refractory, 31% of CR duration less than 12 months, 26% of extranodal disease at relapse) when compared to our series [10,13]. 

The synergistic action of BV with a PD-1 inhibitor seems to rely on the combination of tumor microenvironment modulation and T-cell clonal expansion. Interestingly, if the inhibitory activity towards regulatory T cells has already been described after single-agent BV, the promotion of T-cell clonal expansion after single-agent PD-1 inhibitor has not yet been observed [10]. 

Hence, this combination could lead to a significant improvement for those heavily pretreated patients considered as chemorefractory. In restoring chemosensitivity, some of these patients could be spared from alloSCT, which is burdened with an important treatment-related mortality.

It has been reported in several series that even patients who failed PD1 inhibitor therapy seem to benefit from a re-treatment with chemotherapy, supporting the hypothesis that these drugs can re-sensitize lymphoma cells to conventional treatment after previous failure [14,15]. In a series of 30 R/R HL patients from the LYSA (Lymphoma Study Association) group treated with chemotherapy after unsatisfactory response on PD1 inhibitor treatment, the reported ORR was 67% for the proportion of patients presenting a PD, with a CR rate of 46%. Several patients achieved a response despite being re-exposed to the same chemotherapy agent received before PD1 inhibitor treatment. Furthermore, patients treated with a combination of PD1 inhibitors and chemotherapy showed a trend to a better response than those treated only with chemotherapy, underlining the potential synergy among the two treatments [16]. Similarly, in a retrospective analysis, 81 R/R HL patients from the United States and Canada were treated with several types of salvage chemotherapy after PD1 inhibitors failure: the ORR was 62%, the CR rate of 42% and 47% of patients could proceed to ASCT and/or alloSCT. Interestingly, patients presenting a PD after anti-PD1 treatment were also capable of achieving a response after being treated with subsequent conventional chemotherapy [17].

The potential role of PD-1 inhibitors as chemosensitizing agents has also been studied in the setting of pre-ASCT R/R HL. In a retrospective analysis, heavily pre-treated high-risk HL patients (median of three prior systemic lines of treatment, 62% of primary refractory disease) received a PD-1 inhibitor, as a single agent or in combination with chemotherapy, before ASCT; at a median follow-up of 20 months, the PFS for the entire cohort was 81% with no significant differences among patients presenting a positive or a negative PET-CT (18-month PFS, 75% vs. 85%, *p* = 0.18) [18]. However, it must be underlined that this study did not include a centralized radiologic review and that PET-CT was not assessed using response criteria explicitly designed for immunotherapy in lymphoma, such as LYRIC or RECIL [12,19].

The early introduction of PD1 inhibitors for HL salvage treatment has been explored by Moskowitz and colleagues in a recently published study using a combination of pembrolizumab and GVD (gemcitabine, vinorelbine and liposomal doxorubicin) chemotherapy as first salvage treatment followed by ASCT. This association showed an impressive CMR rate of 95% with no PD reported after 1 year of follow-up [20]. Authors hypothesize that one or more drugs within GVD are capable of enhancing the anti-tumoral activity of pembrolizumab, according to preclinical data which showed that both gemcitabine and doxorubicin present a stimulating activity towards T cell-mediated immunity [21,22]. 

The combination of BV with chemotherapy (bendamustine, GVD, IGEV, ESHAP, DHAP) has also been explored, with encouraging results (CR rates 70–90%); however, it must be underlined that these schemes are markedly more toxic (G3-4 hematological toxicity 50–90%) and seem to be less effective in primary refractory patients [23,24,25,26,27,28,29,30]. 

Our analysis presents limitations due to its retrospective nature, its small size and the absence of biological studies. However, it supports the idea of combining BV and Pd1 inhibitors in relapsed/refractory HL patients. Particularly, we think that this combination is an extremely valid option for HL patients’ refractory to chemotherapy-based salvage treatment.

## 5. Conclusions

Our preliminary data suggest that the BV and pembrolizumab combination is a highly effective and safe bridge treatment to ASCT for high-risk, heavily pretreated R/R HL patients. Compared to other salvage regimens, the efficacy is also confirmed in primary refractory patients. Further studies on larger samples are needed to confirm these data. 

## Figures and Tables

**Figure 1 cancers-14-00982-f001:**
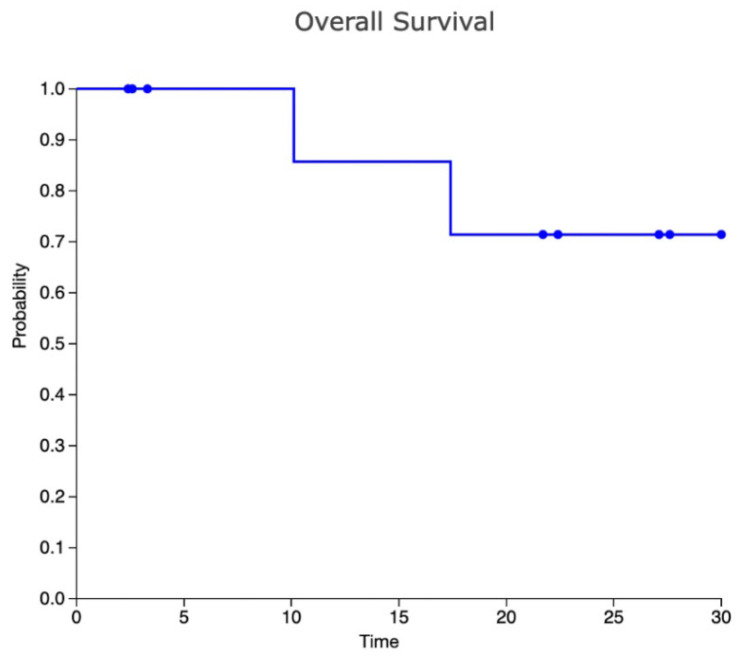
Kaplan–Meier curve for OS.

**Table 1 cancers-14-00982-t001:** Baseline characteristics of patients (*n* = 10).

Characteristics		Patients, *n*.
Median age, years (range)		30.7 (20.6–36.4)
Sex, male/female		8 (80%)/2 (20%)
Median time from diagnosis to pembro-BV, months (range)		27.7 (13.6–51)
First-line treatment	ABVD	5 (50%)
	BEACOPP esc	4 (40%)
	CHOEP	1 (10%)
Number of prior treatments before pembro-BV salvage	2	3 (30%)
	3	6 (60%)
	5	1 (10%)
First salvage therapy	DHAP	6 (60%)
	BEGEV	3 (30%)
	ICE	1 (10%)
Second salvage therapy	BEGEV	2 (20%)
	Bendamustine	2 (20%)
	ICE	1 (10%)
	BEACOPP esc	1 (10%)
Refractory disease		6 (60%)
Complete remission < 12 months		4 (40%)
Extranodal involvement at relapse		6 (60%)
Advanced stage at relapse		9 (90%)

ABVD, doxorubicin, bleomycin, vinblastine, dacarbazine; BEACOPP esc, doxorubicin, cyclophosphamide, etoposide, procarbazine, prednisolone, bleomycin, vincristine; CHOEP, cyclophosphamide, doxorubicin, vincristine, etoposide, prednisolone; BEGEV, bendamustine, gemcitabine, vinorelbine; DHAP, dexamethasone, cytarabine, cisplatin; ICE, ifosfamide, carboplatin, etoposide; BV, brentuximab vedotin.

**Table 2 cancers-14-00982-t002:** Treatment details for each patient in study.

	Prior Lines of Treatment	Pembro + BV (Cycles)	PET2	ASCT	BV Post-ASCT (Cycles)	PD after ASCT	Allo SCT	Follow-Up (Months)	Last Disease Status	Patient Status
Pt1	3	2	DS3	Yes	14	No	No	29.9	CR	Alive
Pt2	2	4	DS3	Yes	5	Yes	Yes a	17.4	CR	Dead
Pt3	3	7	DS4	Yes	9	No	No	27.1	CR	Alive
Pt4	3	2	DS2	Yes	14	No	No	27.6	CR	Alive
Pt5	2	4	DS5	No b	No	Yes	Yes	10.1	PR	Dead
Pt6	2	6	DS1	Yes (+RT)	3	No	Yes c	22.4	CR	Alive
Pt7	3	6	DS2	Yes	1	No	No	21.7	CR	Alive
Pt8	3	4	DS1	Yes	Not yet	NA	No	3.3	CR	Alive
Pt9	5	4	DS1	Not yet	Not yet	NA	No	2.4	CR	Alive
Pt10	3	4	DS2	Not yet	Not yet	NA	No	2.6	CR	Alive

^a^ After subsequent treatment line (nivolumab + gemcitabine) due to PD after ASCT. ^b^ ASCT after subsequent treatment line (nivolumab + gemcitabine and RT) due to PD. ^c^ Directly after ASCT (tandem strategy).

## Data Availability

The data presented in this study are available on request from the corresponding author.

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
