# Peer review of "Brentuximab Vedotin and Pembrolizumab Combination in Patients with Relapsed/Refractory Hodgkin Lymphoma: A Single-Centre Retrospective Analysis"

_cancers, 2022, doi:10.3390/cancers14040982_

Round 1

Reviewer 1 Report

This retrospective analysis describes the treatment and outcomes of 10 patients with refractory HL treated with the combination of BV and pembrolizumab. As the authors note, although these drugs have been studied individually to treat relapsed HL, there are no published studies of the combination making this study of interest. One question is that it looks like none of the patients had received brentuximab or pembrolizumab in any of their prior treatment regimens, would this regimen be recommended it a patient had received prior brentuximab and had progression of disease (not that this can be answered with this data, but is clinically interesting). 

Results: Please clarify, if 60% of patients had primary refractory disease, does the group of patients who had a CR less than 12 months include patients who never achieved a CR? Also the text mentions 60% had primary refractory disease but table 1 says 50% had primary refractory disease

Edits:

Line 41: consider “achieving” instead of “allowing to achieve”

Line 50:

Line 63: was associated with an ORR of 82%

Line 74: and had PET avid measurable disease

Line 104: consider had instead of benefited

Lin 125: with PD instead of a PD

Line 161: consider note instead of notice

Line 206: “are capable of enhancing”

Line 211: candidates

Author Response

Results: Please clarify, if 60% of patients had primary refractory disease, does the group of patients who had a CR less than 12 months include patients who never achieved a CR? Also the text mentions 60% had primary refractory disease but table 1 says 50% had primary refractory disease

Thank you for this precious remark that allowed us to detect an error in reporting clinical data into text. Indeed, 60% of patients had primary refractory disease, 40% of patients a CR less than 12 months, 50% at least 3 high risk features simultaneously. We corrected in the text as explained.

Edits:

Line 41: consider “achieving” instead of “allowing to achieve”

Line 50:

Line 63: was associated with an ORR of 82%

Line 74: and had PET avid measurable disease

Line 104: consider had instead of benefited

Lin 125: with PD instead of a PD

Line 161: consider note instead of notice

Line 206: “are capable of enhancing”

Line 211: candidates

All modifications were done according to your suggestions. However if there's something which has to be changed in line 50 please let us know.

Reviewer 2 Report

As the first publication of which I am aware of the pembro/BV combation, I think this is a very interesting manuscript.  

The authors should clarify the methods re: prospective versus retrospective study. Through out the manuscript they call this a retrospective study. However, in the methods section (2nd and 3rd paragraphs) they say that the treatment proposal was approved by their committee and everyone signed informed consent, which sounds more like a prospective study.

In methods section, around line 80-81 it states that patients will have PET after 2 cycles and then go to ASCT afterwards. Please add a line here to clarify that patients could continue to receive BV-pembro until they went to transplant.  

Line 118 says "The only patien presenting with PD during salvage treatment....".  Is this the same patient who had PD after 6 cycles as mentioned in line 109?  If not, then please clarify.  If it is the same patient then they may not need to present the same data twice especially since they aren't consistent since during the first mention they only discuss radiation while in the 2nd mention they disucss radiation and chemotherapy.  If it is the same patient, I think it would be clearest to the reader to describe the patient in a separate paragraph.  

Line 125 - for the two patient who presented with PD after ASCT, it would be interesting to know how long after transplant their disease progressed.  Also, where these patients in a CMR prior to transplant?

Table 2: 

It seems like the ASCT column should refer to those who got autoSCT right after treatment since that was the main question of the manuscript.  Therefore, consider puttinb No for patient 5 and then have the superscript clarifyling that they eventually got transplant after more salvage treatment. 

Disease status column - does that refer to most recent disease status (for example including patients who are in a CR after treatment for their post-transplant relapse)?  If so, it is probably a little misleading since the table doesn't mention that two patients had relapsed.  Maybe change the column to say ongoing CR after transplant y or no or something like that.  Some sort of clarification is needed.

Median follow up.  Is the data cut off just the last visit of until disease progression. I think the former would be more useful data with regards to efficacy of this regimen.  Median progression free survival woudl be useful information.

Paragraph at 171 seems like a big assumption, especially since 3 patients in this manuscript got alloSCT.

Author Response

As the first publication of which I am aware of the pembro/BV combation, I think this is a very interesting manuscript.  

The authors should clarify the methods re: prospective versus retrospective study. Through out the manuscript they call this a retrospective study. However, in the methods section (2nd and 3rd paragraphs) they say that the treatment proposal was approved by their committee and everyone signed informed consent, which sounds more like a prospective study.

It is indeed a retrospective analysis. We just stated that, as per institutional procedures, patients signed an informed consent and that the treatment was approved at our oncologic multidisciplinary commission.

In methods section, around line 80-81 it states that patients will have PET after 2 cycles and then go to ASCT afterwards. Please add a line here to clarify that patients could continue to receive BV-pembro until they went to transplant.  

We added this statement as suggested.

Line 118 says "The only patien presenting with PD during salvage treatment....".  Is this the same patient who had PD after 6 cycles as mentioned in line 109?  If not, then please clarify.  If it is the same patient then they may not need to present the same data twice especially since they aren't consistent since during the first mention they only discuss radiation while in the 2nd mention they disucss radiation and chemotherapy.  If it is the same patient, I think it would be clearest to the reader to describe the patient in a separate paragraph.  

It is two different patients. We clarified that the patient at line 119 is the only one progressing at the PET-CT evaluation after 2 cycles of treatment.

Line 125 - for the two patient who presented with PD after ASCT, it would be interesting to know how long after transplant their disease progressed.  Also, where these patients in a CMR prior to transplant?

We added these data, as suggested.

Table 2: 

It seems like the ASCT column should refer to those who got autoSCT right after treatment since that was the main question of the manuscript.  Therefore, consider puttinb No for patient 5 and then have the superscript clarifyling that they eventually got transplant after more salvage treatment. 

We modified as suggested.

Disease status column - does that refer to most recent disease status (for example including patients who are in a CR after treatment for their post-transplant relapse)?  If so, it is probably a little misleading since the table doesn't mention that two patients had relapsed.  Maybe change the column to say ongoing CR after transplant y or no or something like that.  Some sort of clarification is needed.

We added the column "PD after ASCT: yes/no" to clarify this aspect.

Median follow up.  Is the data cut off just the last visit of until disease progression. I think the former would be more useful data with regards to efficacy of this regimen.  Median progression free survival woudl be useful information.

Median follow-up is related to last visit. We add median PFS value into the text.

Paragraph at 171 seems like a big assumption, especially since 3 patients in this manuscript got alloSCT.

We agree and we changed this statement in order to "mitigate" its meaning.

Reviewer 3 Report

This is a very interesting report with limitations of a retrospective study. The association of new drugs in RR/HL patients is a clear need. This studies could help to induce local authorities to label the use of these drugs in a very reduced number of patients.

Some points to be clarify:

  • why patient 6 performed allogeneic transplant
  • why patient 1 and patient 3 performed ASCT even if PET was positive, do you coinsider the DS 4 as an effect of PD1 (aspecific positivity?). Should be specify.
  • The discussion is focalized on the resensitizing effect of PD1 to chemotherapy. I think should be better to reduce this part becouse was evaluated in very few patients.
  • Should be interesting to know how many patients in the same period of observation could not be salvaged with this therapy becouse were not covered by special insurance

Author Response

This is a very interesting report with limitations of a retrospective study. The association of new drugs in RR/HL patients is a clear need. This studies could help to induce local authorities to label the use of these drugs in a very reduced number of patients.

Some points to be clarify:

  • why patient 6 performed allogeneic transplant
    patient 6 received allogeneic stem cell transplant because despite presenting a CMR after 2 cycles, he progressed after 6 (ASCT delayed during first wave of covid pandemic). So, given that the patient progressed after both antiCD30 and antiPD1 treatment but was in CMR after ASCT, we decided to further consolidate with allo SCT, estimating the patient at high risk of relapse and with limited treatment options.
  • why patient 1 and patient 3 performed ASCT even if PET was positive, do you coinsider the DS 4 as an effect of PD1 (aspecific positivity?). Should be specify.
    there was a mistake concerning PET-CT result for patient 1, which presented indeed a DS3: we modified the value in the table. Patient 3 presented a partial response after 2 cycles with a DS4, so we decided to administrate further cycles of treatment to improve the response but after 6 cycles he still presented a DS4 with a single pulmonary lesion which we failed to biopsy. So we performed a 7th cycle and we proceeded to ASCT.
  • The discussion is focalized on the resensitizing effect of PD1 to chemotherapy. I think should be better to reduce this part becouse was evaluated in very few patients.
    We shortened the discussion eliminating a few not indispensable lines
  • Should be interesting to know how many patients in the same period of observation could not be salvaged with this therapy becouse were not covered by special insurance
    Interesting question, however clearly hard to evaluate, especially because this is a cohort of heavily pretreated patients which did not received BV as I or II salvage treatment
    Thank you for your comments !

Reviewer 4 Report

In this retrospective study, the authors evaluated the efficacy of a combination of brentuximab vedotin (BV) and pembrolizumab in a small number of HL patients presenting with a high-risk multi-refractory disease. Patients achieving a Deauville score ≤4 proceeded to ASCT consolidation. They concluded that the BV and pembrolizumab combination is a very effective bridge treatment to ASCT for high-risk R/R HL patients. It is an interesting studt

The authors should describe a therapeutic algorithm for Relapsed or Refractory Hodgkin Lymphoma according to their experience.

They should also inform the PET/CT Criteria that have been used  for Prediction of Response to Immune Checkpoint Inhibitor Therapy.

Author Response

In this retrospective study, the authors evaluated the efficacy of a combination of brentuximab vedotin (BV) and pembrolizumab in a small number of HL patients presenting with a high-risk multi-refractory disease. Patients achieving a Deauville score ≤4 proceeded to ASCT consolidation. They concluded that the BV and pembrolizumab combination is a very effective bridge treatment to ASCT for high-risk R/R HL patients. It is an interesting studt

The authors should describe a therapeutic algorithm for Relapsed or Refractory Hodgkin Lymphoma according to their experience.

They should also inform the PET/CT Criteria that have been used  for Prediction of Response to Immune Checkpoint Inhibitor Therapy.

Thank you for your comments.

We added a line at the end of discussion, identifying this combination as extremely valid for patients refractory to chemo-based first salvage treatment.

We also added a line in methods to clarify PET-CT criteria used to evaluate disease response.